# On the Challenges and Potential of Using Barometric Sensors to Track Human Activity

**DOI:** 10.3390/s20236786

**Published:** 2020-11-27

**Authors:** Ajaykumar Manivannan, Wei Chien Benny Chin, Alain Barrat, Roland Bouffanais

**Affiliations:** 1Engineering Product Development, Singapore University of Technology and Design, 8 Somapah Road, Singapore 487372, Singapore; manivannan_ajaykumar@mymail.sutd.edu.sg (A.M.); benny_chin@sutd.edu.sg (W.C.B.C.); 2CNRS, CPT, Aix Marseille University, Université de Toulon, 13009 Marseille, France; alain.barrat@cpt.univ-mrs.fr; 3Tokyo Tech World Research Hub Initiative (WRHI), Tokyo Institute of Technology, Yokohama 226-8503, Japan

**Keywords:** barometer, barometric pressure, human activity recognition (HAR), vertical displacement activity (VDA)

## Abstract

Barometers are among the oldest engineered sensors. Historically, they have been primarily used either as environmental sensors to measure the atmospheric pressure for weather forecasts or as altimeters for aircrafts. With the advent of microelectromechanical system (MEMS)-based barometers and their systematic embedding in smartphones and wearable devices, a vast breadth of new applications for the use of barometers has emerged. For instance, it is now possible to use barometers in conjunction with other sensors to track and identify a wide range of human activity classes. However, the effectiveness of barometers in the growing field of human activity recognition critically hinges on our understanding of the numerous factors affecting the atmospheric pressure, as well as on the properties of the sensor itself—sensitivity, accuracy, variability, etc. This review article thoroughly details all these factors and presents a comprehensive report of the numerous studies dealing with one or more of these factors in the particular framework of human activity tracking and recognition. In addition, we specifically collected some experimental data to illustrate the effects of these factors, which we observed to be in good agreement with the findings in the literature. We conclude this review with some suggestions on some possible future uses of barometric sensors for the specific purpose of tracking human activities.

## 1. Introduction

### 1.1. Development of Barometric Sensors

Barometers have been around for a very long time. While the air was thought to be weightless till the early 1640s, this changed when the Italian physicist and mathematician Evangelista Torricelli showed that a column of air exerts a significant force that can be measured by the amount of liquid displaced by the pressing air. This led to the discovery that “air has weight” and the invention of a measurement device that quantifies the atmospheric pressure [1]. In the latter part of 1640s, Blaise Pascal perfected the experiment and showed the finiteness of air pressure, leading to the hypothesis that the height of the atmosphere itself is finite, and to the proposition that altitude can be measured as proportional to the atmospheric pressure [2]. The SI-derived unit, the Pascal (Pa), is named after Pascal’s contributions to hydrodynamics, and is now officially used to measure the force applied by an atmospheric column of air above a unit surface area.

For the first two centuries after the invention of the barometer, this device was constructed using glass tubes filled with liquids, such as water or mercury [3]. In 1844, a new design appeared with the development of the aneroid barometer, which is purely mechanical, does not contain liquids, and shows the measurement value on a face dial [4]. These devices took a quantum leap with the advent of micro-fabrication in the 1960s, which allowed the miniaturization of the barometer and accelerometer to a size smaller than 0.1 mm—what is now commonly known as a microelectromechanical system (MEMS). With the successive advancements in integrated circuits and digitization of the sensor readings, the manufacturing and computational costs of these miniaturized devices were significantly driven down, thus paving the way for their widespread adoption in consumer products, especially in mobile phones. Currently, MEMS-based barometers are by far the most commonly found type of barometer in wearable devices and smartphones. In 2015, one of the pioneers in MEMS manufacturing, Bosch (Robert Bosch GmbH), claimed to manufacture one billion MEMS sensors per year for automotive and non-automotive applications in one production facility in Germany [5]. This company also claims to have their MEMS devices (including barometers) embedded in every second smartphone in the world [5]. Although this claim cannot be independently verified, it still points to the massive scale of production for this type of sensor, and underscores the ubiquitous availability of MEMS barometers.

Historically, barometers were used for weather forecasting and, thus, chiefly as environmental sensors. As a measurement device of ambient pressure, barometers have recently been used to measure evapotranspiration (transfer of water from land to atmosphere) in a given environment, for improving motor vehicle engine efficiency by modifying the air–fuel mixture, and for counting steps based on the slight disturbances in air pressure during body movements [6]. Thanks to the relation between pressure and altitude, barometers are also widely used as altimeters to measure altitude, particularly in airplanes. A number of recent applications are directed to the tracking of human activities, and there are still significant opportunities for the use of barometers in a vast range of additional applications. The full potential of barometers has not yet been taken advantage of, particularly in the Internet of Things (IoT) realm and with future consumer devices, particularly in the fast-growing area of wearable devices.

### 1.2. Barometers for the Tracking of Human Activity

In particular, the ubiquity of MEMS barometers in smartphones and other wearable devices makes them natural candidates as data sources for the study of human activities and for the field of human activity (and movement) recognition (HAR) [7]. Broadly speaking, HAR consists of using data from various types of sensors carried by individuals to automatically understand what type of activity they are carrying out. It consists, therefore, of choosing sensors that will be influenced by the activity, annotating a certain amount of data (that will serve as “training data”) thanks to ground truth knowledge (i.e., the knowledge of the precise conditions in which the data were collected), and devising a classification task using typical machine learning frameworks to classify the rest of the data. The human activities considered can be broadly classified under two main classes: (1) ambulation and (2) transportation [7]. Ambulation refers to all movements and idle states of our human body (walking, idle, running, sitting, etc.), while transportation refers to our movement using vehicles (cars, buses, bicycles, etc.) [7]. In HAR research, such activities are considered under two conditions: (1) natural ones and (2) laboratory conditions. These conditions lie at the two extremes of a spectrum: “Natural” refers to individuals behaving normally within their usual environment, without any defined procedure and without being influenced by their being monitored, while laboratory conditions refer to a set-up that is especially designed for a human subject, who is given explicit instructions to perform a given activity. In reality, most experiments with human subjects happen somewhere in between these two extreme conditions. It is also important to note that the commonly tracked activity classes vary for healthy and differently abled individuals. For example, tracking patients with movement disorders requires detection of distinct movement features, such as tremors, myoclonus, etc.

Until the 1990s, it seems that there is almost no reference in the literature to the use of barometers in tracking human activities, the most commonly used sensors being inertial measurement units (IMU)—comprising the accelerometer and gyroscope, along with the magnetometer. For instance, smartphones are equipped with accelerometers, whose signals are widely used to recognize most activity classes [8]. The situation changed with the silicon and digital revolution, which contributed to the effective use of barometers to track a range of human activities in the late 1990s. Initially, consumer devices such as mobile phones were equipped with barometers to improve GPS-based localization by reporting altitude or altitude changes [8]. Today, barometers along with a suite of sensors like IMUs and magnetometers are used—individually or through sensor fusion—to track a wide range of human activities. Such use of a barometer to track human activities is a fairly recent phenomenon. In the last two decades, barometers were found to improve some activity class recognition that involves change in height, such as falls or vertical movements. In some applications, such as recognizing vertical displacement activity (VDA), the accelerometer has been replaced by or at least given less importance than barometers, which are more energy efficient, require less signal processing, and yield less noisy signals than IMU signals. Most smartphones have barometers allowing them to predict changes in altitude with an accuracy of the order of one meter. However, an effective tracking of human activities is best obtained with the combined use of these important sensors, each providing unique information on the subject’s state.

It is, however, important to note that our ability to properly leverage the potential of barometers for HAR purposes critically hinges on our understanding of the physical properties of the atmospheric pressure. Indeed, the measure of the ambient pressure by a barometer is influenced by the static and dynamic properties of its environment [9]. Its effective use as a signal thus requires dedicated data post-processing techniques and classifiers in general to account for external factors: For instance, if one is tracking altitude changes and vertical displacement activity, the variations in the local atmospheric pressure have to be accounted for. Measures are moreover affected by the sensor itself, whose accuracy and manufacturing imperfections can introduce noise and variability between devices. Hence, it is fair to say that the numerous factors affecting barometric pressure (see Section 4), if not properly understood and accounted for, can hinder the effective use of barometers to detect and identify particular classes of human activity.

In order to systematically discuss the issues related to the use of barometric pressure to track human activities, we have collected and organized a series of scientific articles to understand the potential and limitations of such an endeavor. The current review is non-exhaustive but intentionally limited to the use of barometric sensors for the most common classes of human activity. This choice is justified by the wide range of applications offered and the fact that barometers now pervade many mobile devices (wearables and smartphones, cars, etc). The applications explored are also primarily limited to the recent developments in using MEMS-based barometers for consumer goods/electronics. It is worth stressing that this review does not address the vast breadth of machine learning (ML) or other advanced classifiers used to interpret the sensed data or to identify a given activity with a given accuracy. However, the details provided in this review are most useful for the further development and design of effective ML/classifier strategies.

This review is organized as follows: Section 2 introduces the general sensor data collection process to track human activity and discusses a number of general issues common to many types of sensors. Section 3 focuses more specifically on the use of barometers in human activity and movement recognition. Section 4 describes the factors that affect barometric pressure and quantifies the order of magnitude of each effect based on a range of studies reported in the literature. This section is enriched by data especially collected for illustration purposes. Lastly, Section 5 reviews the key findings of the above sections, explores the current challenges, and recommends future directions for different applications of barometric sensing applied to tracking a range of human activities.

## 2. Tracking Human Activities: Background about Data Collection

Three components are key to studying and understanding human activities, namely (i) motion tracking, (ii) recognizing the type of activity, and (iii) analyzing the obtained patterns. Barometers are part of the suite of sensors used in each of these steps. As such, the characteristics of the data collection process are similar to those for other types of sensors. They include the characteristics of sensors, their placement and orientation, the sampling frequency, and the environmental conditions. The type of application (activity class, diversity of sample population) and the method for recording the ground truth and annotating a part of the data also have to be carefully designed.

Data collection and annotation are indeed critical to the effectiveness of subsequent stages of classification—e.g., data pre-processing, feature engineering and identification—in the overall workflow of sensed data associated with human activity recognition. Similarly, an informed decision on the data collection procedure depends on the specific problem under study and on the particular class of human activity being investigated. Depending on the sensor characteristics and the class to be recognized, data collection methods must be tailored to shed light on the phenomena under investigation for better accuracy and performance. The success of such data-processing activity not only depends on acquiring the data, but also on being able to effectively process it and extract meaningful features and patterns. In this section, we successively discuss the key elements of data collection to track human activities: the commonly used sensors and sensor suites, the placements and orientation of sensors and their effects, the sampling frequency for tracking and detecting the activity classes of interest, the types of activity classes related to human movements, the nature of data in terms of the levels of realism (natural, laboratory, and in between), the class imbalance issues, the diversity in sampling targets (users), and, lastly, the challenges in annotation techniques.

### 2.1. Sensors and Sensor Suites

Smartphones and watches are by far the most common and straightforward way to collect data in a natural setting. Nonetheless, some research groups studying particular behaviors and phenomena required a custom-built wearable sensor design for both laboratory testing as well as for operations in real-life conditions. For instance, three-axis accelerometers are the most widely used sensors [10,11], followed by three-axis gyroscopes and magnetometers. Since these inertial sensors are now commonly found in today’s smartphones and other MEMS devices, many recent studies use a combination of these sensors to improve classification accuracy [12,13], although a three-axis accelerometer alone can extract good-quality data, resulting in excellent classification results [14]. Wearable devices may also contain environmental sensors that measure, for instance, temperature, light, atmospheric pressure, and sound to assist in context detection, and/or physiological sensors, such as for the heart rate for medical research [7] and personal fitness purposes. In transport mode recognition, GPS is the most widely used sensor [15], followed by telecommunication data [16,17], WiFi access points [18,19], and travel surveys [20]. In general, location-based sensors operate based on a combination of inertial sensors to build robust recognition systems that can distinguish between different travel modes, including walking and being idle [21]. Fusion of environmental sensor data with inertial and location-based sensed data provides sufficient information to detect what is commonly known as ‘Activities of Daily Living’ (ADL). In this context, barometric pressure sensors have traditionally been considered as environmental sensors used to measure ambient pressure. However, they are also capable of sensing movement/activity in ways similar to inertial sensors, especially when considering vertical movements.

### 2.2. Placement and Orientation of Sensors

The placement and orientation of the sensor might influence the characteristics of the captured signal, thereby affecting the recognition accuracy: Indeed, the training data might then fail to account for all the possible variations, often resulting in a sparse feature space. Numerous studies have focused on ways to alleviate this particular effect, and have provided solutions that range from collecting diverse data to independent features not affected by such parameters [22,23,24]. Chen et al. [25] proposed the use of coordinate transformation along with principal component analysis (PCA) to reduce this issue associated with orientation changes. Several groups have also studied the question of optimal sensor placement [26,27]. Interestingly, while this issue affects many sensors, this is not the case for barometric pressure sensors, whose readings are widely independent of their on-body position and orientation [28], even if they are dependent on a range of environmental conditions.

### 2.3. Sampling Frequency for Capturing Human Activities

The temporal resolution of the data is directly related to the sampling frequency of the sensor used. For the vast majority of sensors used to carry out human activity recognition, the sampling frequency typically ranges from 10 to 100 Hz, with the rate going as high as 512 Hz. It is commonly reported that the characteristic frequencies of most human activities are below 10 Hz, and therefore, the optimal sampling rate—based on the Shannon–Nyquist theorem—is 20 Hz [29]. Khan et al. [30] reviewed five public datasets and showed that the sampling rate considered could be reduced between 48% and 86%, with a minimal sampling rate of approximately 12 Hz [30]. Yan et al. [31] also studied the effect of sampling rate on energy consumption, and they concluded that a higher sampling rate increases energy load without providing additional meaningful information, in agreement with the results reported in [30]. From the energy perspective, Yan et al. [31] concluded that a shorter dataset with a higher sampling rate is preferable to a longer dataset at a lower sampling rate. It is worth adding that the optimal sampling rate also depends on the type of human activity to be recognized [30] and on the type of sensor used. For instance, when studying large-scale human movement, sampling GPS data or other location signals at rates comparable to those of accelerometers is unnecessary. Hence, the GPS signals are typically sampled at around 1 Hz [32,33]. For pressure signals, barometers have been sampled at rates as low as 1 Hz [34].

### 2.4. Classes of Human Activity

Lara and Labrador [7] thoroughly reviewed a comprehensive list of human activities recognized and categorized in the literature, including ambulation, transportation, gestures, exercises, and daily living activities. Ambulation refers to all states of movements performed directly by a human, e.g., walking, idle, running, and sitting. Transportation refers instead to one’s movement when riding vehicles, including cars, buses, and bicycles. Gestures also correspond to body movements, such as picking up or putting down objects, looking at watches, moving the head, etc. Exercises comprise the motion of the human body for the purpose of physical health and activity (jogging, running, playing ball, etc.). Daily living activities are a group of activities that most people experience on a daily basis (e.g. eating, drinking, and reading). The activity classes may, of course, occur concomitantly and may, therefore, be composite [35], interleaved, concurrent, or overlapping [36]. The design of data collection—e.g., the selections of sensors, sampling rates, and recognition approaches—is heavily dependent on the types of activity classes to be recognized. For data with low sampling frequency, i.e., lower than 1 Hz, more than one activity could be performed during the same time interval. In particular, the class of vertical displacement activities (VDA) has generally been recognized in the literature as part of the larger class of ambulation activities. Similarly to accelerometers, barometers are very well suited for the recognition of VDA. In addition, barometers can accurately determine altitude changes for VDA occurring at a sampling rate lower than 1 Hz.

### 2.5. Nature of Data

The data used to study human activity can be broadly grouped under two extreme conditions: (1) natural and (2) laboratory. While the former concerns data collected during real-life activities of people in a given city or country, and might thus be affected by complex environmental factors and all kinds of unknown conditions, the latter corresponds to activities performed in fully controlled environments with known conditions. Several groups have collected their own datasets to fit the specific needs and requirements of their studies [37,38]. Many benchmark datasets are also publicly available, and are commonly used to validate new methods [39,40,41,42]. The recent review article [43] reported details of several key benchmark public datasets and provided a rich analysis on the content and application-context studies. The datasets collected and made publicly available were either collected in laboratory conditions [44] or are real-world ones—where participants had instructions to perform specific activities given implicitly [45] or explicitly [24], or were instead allowed to move freely while being unobtrusively observed [13]. These datasets have been claimed to be semi-naturalistic or realistic even when the subjects were given scripted instructions [46,47]. Lara and Labrador [48] reported on subjects performing interleaving activities as they naturally occur, in sequence rather than segmented, to reach realistic conditions. Truly realistic data correspond to data collected while people go about their activities of daily living (ADL) [14]. As highlighted in [49], the use of purely realistic data would, however, require a long data collection period. Hence, segmented data obtained by a set of experiments corresponding to various activity classes are typically collected for training. Some studies such as [50,51] have managed to deal with this issue by considering segmented data for training and activities of daily living or sequence data for validation.

### 2.6. The Class Imbalance Issues in the Tracking of Human Activity

The problem of class imbalance in datasets—the fact that some activities are more prevalent than others and are, therefore, over-represented—is a recurrent condition found in both public and private datasets and known to affect recognition accuracy [47,52,53]. For instance, data corresponding to physically tiring activities, or that are difficult to obtain for other reasons, naturally form an under-represented portion of the data, thereby leading to inherent data imbalance [38]. This issue can be offset by oversampling the minority class or under-sampling the majority class. Chen and Shen [38] used the so-called synthetic minority over-sampling technique (SMOTE) for oversampling the minority class, while Nguyen [54] used a modified version of it. Guan et al. [52] also proposed the use of an ensemble of deep learning models to offset such a data imbalance. Other studies have tackled the problem by simply using F1-score to report classification accuracy that takes into factor the different sizes of each class [52,55]. This issue can occur in applications using barometers, as the number of altitude changes in human movement for any given continuous experiment is usually very limited, and these changes typically have, moreover, a rather high inter-event time in ADL.

### 2.7. The Physical Characteristics of Carriers

The physical characteristics of the sampled users/carriers directly affect the measured human activity patterns; thus, they are an important aspect of the data collection process for HAR. Data collection methods can contribute to the generalization in recognition models by allowing diverse characteristics or parameters to be incorporated into the training dataset. These characteristics should be representative of the final dataset to which the model is applied. One of the important parameters is the user themself, whose diverse physical characteristics can be challenging to integrate or model [37]. Studies generally report the user’s age, height, weight, body mass index (BMI), physical ailments, etc. for context and applicability of the study [55,56,57,58]. In the case of barometers, the height of the user/carrier has only a limited effect on the use of a barometer, and the resolutions of most wearable sensors are within the range of placement difference. It is thus safe to assume that the barometer sensor works independently of the physical characteristics of the carrier.

### 2.8. Annotation Techniques

Stikic et al. [59] note that the accuracy of annotation is subject to a trade-off between length of the data collection and the time and effort required for labeling the dataset. Indeed, if direct observation is required for accuracy, this results in a prohibitively expensive requirement. De la Hoz Franco et al. [43] carried out a meta-review of 374 papers on human activity recognition (HAR) and found that 60% of the used data were annotated. However, most studies record ground truth by resorting to experiments in a laboratory setup with the help of researchers and supporting infrastructure [27,57,58]. Some studies have also relied on subject self-reporting [37,60,61], which is known to be error-prone due to the obvious difficulty in marking precise times while carrying out the activities of daily living. Chung et al. [13] recorded ground truth by unobtrusively following the subjects one at a time. Video capturing [34,47,55,62], audio recording [63], and GPS localization have been highlighted as methods of direct observation without requiring one to monitor the subjects under constrained and controlled conditions. Recording ground-truth data for long-term studies of ADL has been repeatedly acknowledged to be impractical and/or prohibitively manpower intensive [49,64]. For instance, Willetts et al. [14] used automatic cameras to record ground truth every 20 s for 143 participants over 24 h. This shows the interest of automated methods, such as the active learning methods developed by Bota et al. [50], which require the manual annotation of a small subset of the data, while the rest is automatically labeled.

A practical, privacy-sensitive, and unobtrusive annotation method for studies using barometers to track humans over long durations is manual labeling of the data. Unlike the signal sensed by accelerometers or other inertial sensors, barometric pressure can indeed be less complex to interpret. However, this process comes with several challenges, one of which is the lack of complete understanding of all the factors that influence barometric pressure. This important issue is addressed in detail in Section 4.

## 3. On the Use of the Barometer in Human Activity Recognition

For the sake of studying human activity recognition, the barometer has been primarily used to measure changes in altitude (or elevation). The scale of the altitude changes varies from a fall to vertical displacements like moving uphill, climbing a deck of stairs, riding an elevator, etc. In recent applications, the patterns of the time series of barometric signals have been shown to be a good indicator of the underlying activities, such as walking, being idle, and using transportation. The rate of pressure change can help to identify the mode of vertical transport and determine the vertical velocity of air vehicles. Barometers are widely embedded in wearable devices and used for vertical transport detection [28,65,66,67,68,69,70], indoor positioning and navigation [68,71,72,73], building monitoring [74], health monitoring [75,76], vehicle tracking [77,78], transport mode detection [79], and GPS localization improvement [68].

One of the earliest works on using barometers to classify vertical movement is due to Sagawa et al. [80] and dates back to 1998. In [80], Sagawa et al. collected 83 min of data using both an accelerometer and barometer at a 100 Hz sampling frequency from six males between 20 and 40 years of age. Their classification model was trained offline using cut-off values selected heuristically. Since then, recognition and identification procedures have greatly improved thanks to a number of factors, including new sensor types, larger range of sampling rates, user and device characteristics, modes of carriage, power consumption, real-time demand, classification models, and, finally, ground truth availability.

### 3.1. Barometric Pressure Sensor for Tracking Human Activity

MEMS barometers are miniature sensors (<0.1 mm) manufactured by prominent companies, such as Bosch [77,81,82], ST Microelectronics [77,81,83], and Measurement Specialities [84]. As mentioned previously, they are commonly found in wearable devices and smartphones. They have high precision but relatively lower absolute accuracy than table-top barometers. Specifically, the accuracy in measuring absolute pressure is low in these sensors, but their relative pressure accuracy is shown to be as low as ±1.2 Pa [84,85], with average mobile devices having a resolution of ±12 Pa [28]. It is worth adding that the measured absolute pressure varies from one device to another due to manufacturing differences and other technical factors [81].

### 3.2. Data Processing for Sensed Barometric Pressure

Barometers are constrained by their resolution and sampling frequency, which limit precision and accuracy, and the use of their data is further constrained by the presence of noise. Some studies have used filtering and signal modeling to overcome the latter problem. For instance, moving average filters over a given time window is widely used for that purpose [71,84,86], followed by other finite impulse response (FIR) filters [87] and infinite impulse response (IIR) filters [76,87,88,89], such as double exponential smoothing [28,69]. Signal modeling, like the sinusoidal fitting model [90] and sigmoidal nonlinear fitting, is commonly used to increase the contrast of elevation changes [91].

The time series data of barometric pressure are generally converted into statistical [70,92], spectral [70,92], temporal [70,92], or wavelet-based features [75,93] before being fed into a classifier. These features are designed to enhance the detection of the specific activity of interest. Most of the features used are based on the rate of change of pressure [69,70,79,84,94] (also known as vertical velocity, or simply the slope) and the standard deviation of differential pressure (dp) [74,79], which differentiates altitude changes from other environmental factors that influence ambient pressure.

### 3.3. Classifiers for HAR

The choice of the classifier depends on the application at hand. It also depends on the range of pressure variations and durations under consideration. Some studies have only considered indoor pressure profiles of individuals [28,68], while few have considered the full range of activities of daily living (ADL), including both indoor and outdoor events like transportation [77,79]. A person carrying a barometer is indeed affected by factors present outside their specific environment, which seriously limits the scope of many studies of HAR using barometer data.

Barometric pressure is more straightforward than inertial sensors in conveying sensed information due to its fairly direct reading, which greatly simplifies the use of classifiers. The most widely used classifiers are decision trees [28,65,68,82,83,87], support vector machines (SVMs) [75,77,95,96], and threshold-based models [76,91,97,98]. Clustering models, such as hierarchical clustering [73,99] and *k*-Means clustering [71], Bayesian-based classifiers [72,100,101], LSTM models [28,102], and fuzzy inference models [90] have also been used. Hidden Markov models take advantage of logical activity sequences that can be associated with some activity sequences (such as riding an elevator before and after walking) [103], while fuzzy inference models take advantage of context and behavioral constraints [90].

Liu et al. [69] compared classifiers such as Random Forest, J48 decision trees, artificial neural networks (ANNs), SVMs, and Naïve Bayes to classify horizontal displacement activity from vertical displacement activity; the Random Forest classifier was found to have the highest accuracy. Vanini et al. [28] compared the performance of Bayesian networks, decision trees, and recurrent neural network (RNN) models to recognize VDA. It has been found that RNNs have a 99% accuracy, while decision trees provide the optimal trade-off in terms of computational cost, energy efficiency, and accuracy.

Some applications do not require any classifiers. For instance, state estimation, like altitude and vertical velocity, can be determined using a Kálmán filter or one of its many variants [80,86,96,104,105]. As another example, Bollmeyer et al. [84] solely used the differential pressure to estimate altitude. Ho et al. [78] used the so-called dynamic time warping (DTW) technique to establish the correlations between the pressure time series data and known geographical elevations to track vehicles. Similarly, Hyuga et al. [106] found ways to use the variance of differential pressure to account for the variations in barometric pressure associated with air velocity and the built environment; they used similarity measures between pressure and known altitude to locate a train/user in a rail network.

### 3.4. Applications: Barometer-Only Studies

Very few studies dedicated to tracking human activity and/or movement solely rely on the use of a barometer, despite the fact that their potential for applications is promising. The selected applications of using only a barometer for HAR are summarized in Table 1, in which we highlight the role of barometers in each study, the considered factors that may affect the performance of barometers, and the key findings of the studies. In addition, the types of human activity (i.e., the activity classes: (A) ambulation and/or (T) transportation), the built environment (i.e., locations: indoor and/or outdoor), and the time period of activities (short term and/or long term) were categorized for comparison purposes. These applications were selected based on three criteria: (1) They use only barometers as sensors, (2) they are representative of their field/sub-field or add unique methodology or results in their field, and (3) they add to the understanding of the many factors that influence barometric pressure. Most of the studies focused on ambulation activity (10 out of 16), while two of them focused on transportation activity, two of them studied both, and the rest (2) were neither about ambulation nor transportation activities. Among the 16 studies, nine were performed in an indoor environment, three in the outdoors, and the other four considered both indoor and outdoor environments and the corresponding factors. Most applications (11 out of 16) focused on the changes in barometers’ readings in a short-term period, two studies focused on long-term patterns, and the other three considered both short- and long-term evolution. Based on these 16 studies, the main factors that may affect the performance of barometers included the altitude of the activity, the background climate and weather, the surrounding built environment, the relative air velocity due to motion, and the sensor accuracy. We discuss the results of these studies in more detail in the following paragraphs.

Barometers have been used to measure altitude as a stand-alone measuring instrument for a very long time [81]. The surface of the land on which we move and travel, including transportation routes, is uneven. This topographical feature can be estimated using barometric pressure, and the subtle changes in elevation along the travel routes can be exploited for localization of people riding vehicles. As stated before, this is, however, challenging due to two major factors that affect barometric pressure—built environment, like tunnels, bridges, etc., and the air velocity during motion. However, Hyuga et al. [106] used the pauses of trains in stations between successive train rides to locate a user/train in a subway route by computing the successive altitude changes and comparing them with known relative elevations of train stations.

Similarly to accelerometer data, the signal pattern encoded in a barometer output carries sufficient information to recognize a range of human activities. Ghimire et al. [87] observed the change in air pressure when a person walks with hands swinging and used this gait pattern to count steps. The gait pattern is also used to detect the walking class with approximately 95% accuracy [87], similar to the performance of available accelerometer-based recognition methods [63,107], and this accuracy fluctuates for both sensors based on the on-body sensor position. In this application, the barometric data will also be person-dependent like with an accelerometer, since the gait patterns detected from air pressure changes can vary among the population. The pressure fluctuations due to vehicle motion were used by Sankaran et al. [79] to differentiate between the distinct patterns produced by people riding vehicles as opposed to walking or standing idly.

In climate-controlled buildings, changes in barometric pressure can be detected during indoor-to-outdoor transitions. Wu et al. [74], moreover, showed the possibility to detect the opening or closing of a building’s entrance doors, even with a barometer located far from the doors, and even analyzed the patterns to determine the type of door (automatic or manual). They highlighted implications for building monitoring and security, along with the potential application of tracking human movement in a building.
sensors-20-06786-t001_Table 1Table 1Categorized literature related to the application or use of barometers for human activity recognition. Two types of activity classes: (A) ambulation and (T) transportation.Ref.Use of BarometerFactors ConsideredContributions and ApplicationsActivity Class(es)LocationTime Period[81]Estimate altitude and altitude changesAltitude, climate and weather, and sensor accuracyEvaluate sensors to estimate the altitude of airplane above ground and the orientation angle of wings using dual-device systems–IndoorShort[106]Estimate altitude and altitude changesAltitude, air velocity during motion, and built environmentEstimate the location of a traveler in a subway using only a barometerTOutdoorShort[87]Detect gait patterns and estimate altitude changesAltitudeStep detection and activity recognition including VDA using a barometerAIndoorShort[79]Detect vehicle patterns and altitude changesAltitude, climate and weather, air velocity during motion, and sensor accuracyIdentify transportation modes and ambulation activities using a barometerA & TIndoor and outdoorShort[74]Detect door opening/closing in building and estimate altitude changesAltitude, climate and weather, and built environmentDetect door opening/closing to monitor building activities and recognize VDAAIndoor and outdoorShort & long[108]Estimate altitude and altitude changesAltitude, climate and weather, built environment, air velocity during motion, and measurement accuracyRecommendations to build indoor localization from reference pressureAIndoorLong[66]Estimate altitude and altitude changesAltitude, climate and weather, built environment, and sensor accuracyFloor localization using reference pressure from multiple barometers in each floorAIndoorShort[99]Estimate altitude and altitude changesAltitude, climate and weather, and sensor accuracyCalibration of wearable barometers using crowd-sourcing to enable floor localization. No knowledge of building or additional infrastructure is requiredAIndoorLong[84]Estimate altitude and altitude changesAltitude, built environment, and sensor accuracyStudied the different factors that affect barometric pressure in the built environment. Estimate indoor altitudeAIndoorShort[68]Estimate altitude changes and mode of vertical transportationAltitude, climate and weather, built environment, and sensor accuracyIdentify VDA and mode of vertical transportAIndoorShort and long[104]Estimate altitudeAltitude, climate and weather, and sensor accuracyEstimation of altitude for indoors and outdoorsAIndoor and outdoorShort[28]Estimate altitude changesAltitude, climate and weather, built environment, and sensor accuracyActivity recognition including VDA using only barometer and comparison with accelerometer-only and GPS-only approachesA & TIndoor and outdoorShort[85]Estimate altitudeAltitude, climate and weather, and sensor accuracyBarometer measurement error modeling and correction to track air vehicle–OutdoorShort[78]Estimate altitude and altitude changesAltitude, climate and weather, built environment, air velocity due to motion, and sensor accuracyCompared barometric pressure data with topographical elevation data to localize and track vehiclesTOutdoorShort and long

Applications that use only barometer data are more common in the studies concerned with floor localization and recognition of VDA. In the absence of location sensors, the challenge with floor localization is to have a reference pressure and associate it with the data measured by the considered wearable devices at the moment they enter a given building. To obtain such a reference pressure, Li [108] recommends receiving it from a location that is similar to the environment in which it is deployed; this setup is important, as the reference pressure obtained from reference stations can potentially experience different environmental effects. Xia et al. [66] showed that the barometric pressure pattern can change from floor to floor in idle settings, and hence, installed a calibrated barometer in each floor to collect multiple reference pressures. Ye et al. [99] applied an infrastructure-independent approach by constructing an encounter network—determined by comparing simultaneous pressure changes—and use a root node to calibrate all the mobile sensors. This method is, of course, prone to errors, and the lack of calibration has been shown to result in an accuracy of only 70%.

The challenge for recognizing VDA using only barometer data is that the sensor data should ideally be free from all factors that affect barometric pressure other than altitude. This can be guaranteed if the sensor data are collected from a controlled environment where no other factors that can be mistaken for VDA occur. As shown by Bollemeyer et al. [84], Ghimire et al. [87], and Muralidharan et al. [68], this can be achieved by limiting the studies to indoors, where only weather, sensor accuracy, and built environment effects impact the barometric readings. Liu et al. [104] collected experimental data from outdoors, like mountain climbing, while avoiding activities that cause adverse pressure gradients, like transportation. Vanini et al. [28] demonstrated their VDA recognition capability while considering ambulation and transportation, with, however, a limited set of activities, including, in particular, only cable-cars as an outdoor transportation mode.

In summary, in a specific environment and considering selected modes of activity, the recognition performance of VDA using barometer can be very high. More generally, in any given environment, barometers are shown to perform far better or similarly to other sensors when detecting VDA, and represent the only viable way to extract the magnitude of vertical displacement. Vanini et al. [28] report that with barometer data, one achieves similar performance in classifying VDA compared to with an accelerometer or GPS, but a superior one in energy efficiency and independence in terms of sensor location and orientation. Muralidharan et al. [68] similarly showed that barometer-only classification is significantly more accurate (99%) in recognizing modes of VDA compared to accelerometer-based classification (85%). However, the accuracy for accelerometer-based classification drops below 30% when the mobile phone is used for taking calls or playing games.

### 3.5. Applications: Multi-Sensor Studies

Few studies have employed a barometer as the sole sensor in their application. It is usually integrated with other sensors, like inertial sensors [69,82], environmental sensors (light, temperature, sound, etc.) [109,110], location-based sensors (GPS) [71,96], and communication infrastructure (WiFi, Bluetooth, RFID, etc.) [72,73,100,110]. Several studies are dedicated to improving the sensor fusion of inertial sensors with barometers, which constitutes a critical step in the optimization of activity recognition [86,110,111].

To summarize and categorize the vast breadth of applications of barometers for HAR with multi-sensors, we have gathered some selected studies into Table 2. The studies were selected based on the same criteria as in Section 3.4. Table 2 contains a list of (29) studies that include barometers as part of their multi-sensor set-up. We detail in the table which other sensors are used, the particular role of barometers in the studies, the factors that may affect the barometric readings, and the key contributions of the studies. Similarly to Table 1, we also categorized the studies by activity classes (A: ambulation, T: transportation), built environment (location, including indoors and/or outdoors), and periods of the experiment (time period, including short term and/or long term) for comparison purposes. Most of the studies used barometers together with accelerometers (23 out of 29, 10 of which were using only these two types of sensors), gyroscopes (12 out of 29), and magnetometers (7 out of 29). Other sensors, such as WiFi fingerprinting (six studies), GPS (two studies), and Bluetooth (one study), were used for indoor or outdoor positioning; in addition, light and foot pressure—both were used in one of the 29 studies—were also included for some specific aims. Most of the applications focused on ambulation activities (26 applications), one of the 26 included cycling activity, and another one included transportation activity; only one of the other three studies focused solely on transportation, and the other two could not be categorized. Almost all of the listed applications were conducted in an indoor environment (27 studies), six of which were also tested in an outdoor environment; one study was conducted only outdoors, and in another one, the context was not mentioned. In terms of duration, most studies focused on short-term activity (28 studies), three of which studied both short-term and long-term human activities; only one application focused solely on long-term activity.

In most classical HAR analyses, the classification accuracy in detecting VDA based on sensory data without pressure is usually low [89]; this is not seen as a critical issue, since VDA is not the focus or priority. Increasingly, the barometer has been recognized as an important sensor in HAR, where accurate recognition of VDA is critical to many applications [105]. Hence, a majority of applications that aim to measure altitude or track altitude changes employ a barometer as part of their sensory suite. For instance, in health monitoring applications, the inclusion of the barometer has been shown to improve VDA recognition [44,75,95], fall detection [76,83,93], and estimation of energy expenditure and physical activity [44,91,97,112].

Accelerometers are still the predominantly used sensors in HAR, and have been widely used as stand-alone sensors in recognizing many activities of daily living. They complement barometer-based recognition algorithms in detecting ambulatory movements such as walking, and their use also helps distinguish stair-climbing from other modes of vertical transportation, like using an elevator and escalator [69,101]—and even elevator from escalator [67]. Sankaran et al. [79], however, point to the high cost associated with the use of accelerometers: demands in data acquisition (position and orientation dependent), high sampling rate, complex processing, and classification training.

For indoor localization and navigation applications, obtaining a reference pressure to calibrate all mobile sensors is critical for the system to work. This is more easily obtained in multi-sensor applications. Pipelidis et al. [71] used light sensors to detect the transition between indoors and outdoors so as to derive a reference pressure at ground level, which subsequently serves the detection of floor levels. Communication infrastructures like WiFi, Bluetooth, and RFID are also used to provide additional location information to help assist indoor localization or transmit location-specific data, such as reference pressures in a floor, to assist barometers for calibration purposes. Tachikawa et al. [110] even combined WiFi signal with a microphone and other inertial sensory data to detect the type of indoor location—restroom, desk, elevator, etc.

Barometers are known to speed up the GPS localization through their altitude estimation [68]. Conversely, the altitude information can be accessed from GPS localization [104]. Furthermore, GPS or any location information can help distinguish the transportation modes from ambulation, where the changes in barometric pressure can easily be misunderstood for altitude changes. Even though the elevation changes are present in our transportation paths, the altitude estimation from barometric pressure due to air velocity during motion can be predominant. Some studies have used the barometric pressure instead of GPS to track people riding vehicles [44,77], but this can be very misleading, as the significant changes in pressure due to vehicle motion and the built environment like tunnels and bridges have not been fully taken into account or even properly understood.
sensors-20-06786-t002_Table 2Table 2Categorized literature related to the application or use of multi-sensors with barometers for human activity recognition. Two types of activity classes: (A) ambulation and (T) transportation.Ref.Additional SensorsUse of BarometerFactors ConsideredContributions and ApplicationsActivity Class(es)LocationTime Period[69]Accelerometer, Magnetometer, GyroscopeEstimate altitude changesAltitude, climate and weather, built environment, and sensor accuracyImproved recognition of VDA using barometerAIndoorShort[82]Accelerometer, GyroscopeEstimate altitude changesAltitudeIdentify ambulation activities including VDAAIndoorShort[110]Accelerometer, Gyroscope, Magnetometer, WiFi, MicrophoneEstimate altitude changesAltitudeDetermine location semantics, such as restroom, desk, elevator, etc., using sensor fusionAIndoor and outdoorShort[71]Light, GPSEstimate altitude and floor levelAltitudeVertical indoor mappingAIndoor and outdoorShort[72]WiFiEstimate altitude and altitude changesAltitude, climate and weather, and sensor accuracyImproved barometer measurement error model and sensor fusion for floor localizationAIndoorShort and long[73]WiFiEstimate altitudeAltitude and sensor accuracyImproved floor localization from crowd sourcing using few devices equipped with a barometer–IndoorShort[100]WiFiEstimate altitude and altitude changesAltitude, climate and weather, and sensor accuracyFloor level identification by hybrid approach between barometer-only and WiFi-only methods. The barometer-only approach uses crowd-sensed barometer data for self-calibration and builds an elevation map independently in each deviceAIndoorShort[111]Accelerometer, GyroscopeEstimate altitude and correct accelerometer errorsAltitudeImproved IMU–barometer sensor fusion–IndoorShort[86]Accelerometer, GyroscopeEstimate altitude changesAltitude, climate, and weatherImproved sensor fusion to track vertical motionsAIndoorShort[89]Accelerometer, GyroscopeEstimate altitude changesAltitudeImproved accuracy in recognizing ambulation activities including VDAAIndoorShort[44]AccelerometerEstimate altitude changesAltitudeUsing a barometer improved overall activity recognition including VDA and better estimated of energy expenditureAIndoorShort[95]AccelerometerEstimate altitude and altitude changesAltitudeChild activity recognition including VDA to prevent injuriesAIndoorShort[75]Accelerometer, Magnetometer, Gyroscope
Altitude, built environment, and sensor accuracyActivity recognition including VDA for health monitoring of stroke patientsAIndoor and outdoorShort[93]AccelerometerEstimate altitude changesAltitudeImproved accuracy by including a barometer for fall detectionAIndoorShort[76]AccelerometerEstimate altitude changesAltitudeImproved fall detection using a barometerA and TIndoorShort[83]AccelerometerEstimate altitude changesAltitudeLow-power fall detection for long-term monitoringAIndoor and outdoorShort and long[91]Accelerometer, Gyroscope, Magnetometer, Foot pressureEstimate altitude changeAltitudeActivity recognition including VDA for better estimation of elderly physical activityAIndoorShort[97]AccelerometerEstimate altitude changesAltitude and sensor accuracyDetection of VDA improved the estimation of physical activity and energy expenditureA + cyclingIndoor and outdoorShort[112]AccelerometerEstimate altitude changesAltitude, climate and weather, and sensor accuracyDetection of VDA improved the estimation of energy consumption and physical activityAIndoor and outdoorShort and long[101]AccelerometerCalculate vertical displacement and estimate floor levelAltitude, climate and weather, and built environmentFloor localizationAIndoorShort[67]Accelerometer, GyroscopeEstimate altitude changesAltitudeImproved identification of VDA for indoor localizationAIndoorShort[77]GPSDetect vehicle patternsAltitude, climate and weather, built environment, and air velocity during motionUse the effect of elevation changes in roads and air velocity due to motion to detect the motion state of a vehicle and help identify traffic congestionTOutdoorLong[92]AccelerometerEstimate altitude changesAltitudeReduced position and orientation dependency using a barometerA–Short[80]AccelerometerEstimate altitude changesAltitude, climate and weather, and sensor accuracyEarliest known activity classification including VDA using a barometerAIndoorShort[65]Accelerometer, MagnetometerEstimate altitude changesAltitude, built environment, and sensor accuracyIdentify mode of vertical transportation for indoor navigationAIndoorShort[90]Accelerometer, GyroscopeEstimate altitude changesAltitude and climate and weatherImproved activity recognition including VDA using a barometer by considering behavioral rules and applying context in a two-step processAIndoorShort[113]WiFi, BluetoothEstimate altitude and altitude changesAltitude, climate and weather, and sensor accuracyUsed WiFi/Bluetooth beacon to localize the user and recorded the respective pressure as the reference pressure for that floor. Any change in elevation is then used to identify the floor levelAIndoorShort[114]Accelerometer, Gyroscope, Magnetometer, WiFiEstimate altitude and altitude changesAltitude3D indoor localizationAIndoorShort[70]Accelerometer, Gyroscope, MagnetometerEstimate altitude changesAltitudeIdentify the mode of vertical transportAIndoorShort

## 4. Factors Affecting Barometric Pressure

Atmospheric pressure is the force applied per unit area by a column of air above a specified area [115]. It is caused by the gravitational pull of air molecules to the Earth’s surface. Atmosphericpressure measured by a barometer is also called the barometric pressure. The fact that the atmospheric pressure varies monotonically with altitude has made it a useful measurement tool for indicating heights, and the development of portable barometers has extended the usage of barometric pressure to the indication of the carrier’s position in the vertical dimension, i.e., using the variations in the local atmospheric pressure—a key component in the positioning of users and HAR. However, since this is a relatively complex usage of barometers compared to the measurement of Earth surface elevation, this local atmospheric pressure could be affected by numerous factors (see Table 3).

From a fundamental point of view, these factors include the atmosphere’s fluid properties, such as density, moisture content, temperature, and motion. Speaking more generally, we can attribute the change in local barometric pressure to more abstract factors, such as climate and weather [116,117,118], air velocity during motion, altitude, and built environment, with intricate interplays between all these factors. The magnitudes of the resultant effects vary and the corresponding ranges can be quantified for specific circumstances. Several studies have attempted to quantify these effects and provide a rich understanding of the factors that influence barometric pressure. Based on those studies, the magnitude of changes and the corresponding timescales for the factors influencing barometric pressure are summarized in Figure 1.
sensors-20-06786-t003_Table 3Table 3Literature related to factors affecting barometric pressure.FactorsReferencesClimate and weather[28,68,69,78,79,85,99,100,104,108,116,117,118]Built environment[66,68,74,84,101,119,120,121]Air velocity during motion[77,78,79,106,122,123,124]Sensor accuracy[66,69,72,76,78,85,90,99,100,108,113]

In order to understand and demonstrate the effects of the numerous factors that may affect the barometric pressures, we collected a series of experimental data using a custom-made device embedded with a barometer sensor (BMP280). The device was designed specifically for national-scale human activity data collection and analysis [125]. In addition to this device, we also used two mobile phones (Iphone 7 and Iphone XR) with the same mobile application (Physics Toolbox Sensor Suite App version 1.3.5) and barometer model (BMP280) in some of the demonstrations for comparison purposes. The results of the collected data are presented in the next subsections.

### 4.1. Climate and Weather

The atmospheric pressure distribution on the planet is caused by the differential heating of the sun at different latitudes, which varies from the tropics to the polar regions [116]. The Earth’s tilt also contributes to the heating difference, as well as the nature of the surface being heated, i.e., ocean or land [116]. Moisture content in the air greatly affects the pressure distribution, as the dry and moist air are heated differently. The differential heating hence produces high- and low-pressure regions on the planet. Due to seasonal differences in heating, the corresponding pressure distribution also changes seasonally [116].

The pressure distribution creates isobaric contours—i.e., the points along which atmospheric pressure is constant [116]. By a combination of the pressure differences in these isobaric regions and the Coriolis force generated by the Earth’s rotation, the air is moved from one place to another, creating wind. The resulting forces impact various scales, creating both the long-term climatic system of the planet and the short-term weather patterns observed locally.

It is very important to note that all weather patterns observed are caused by very small changes in pressure—i.e., the maximum change in the sea-level atmospheric pressure never exceeds 5% of the absolute atmospheric pressure [116]. A gentle breeze (15 km/hr) creates a pressure difference of 10 Pa and a strong breeze (45 km/hr) creates a 100 Pa difference [126]. Although the magnitude of the wind factor is significant, their time scales are often slower than the scale of the human activity to be predicted. During the estimation of floor height, Liu et al. [104] showed that a windy day produces error in their estimation of floor height, while Sankaran et al. [79] found that a windy day produced no significant change in their prediction of movement. Sanakaran et al. [79] further noted that neither wind nor rain had any significant impact on HAR. Similarly, Vanini et al. [28], whileperforming their VDA classification, found that neither cloudy nor rainy weather had any impact whatsoever. Bao et al. [85] estimated the error caused by the wind and developed a model that takes into account the dynamic pressure change to remove this effect based on the wind speed. Figure 2a shows an example of the effect of wind on barometer data: The pressure fluctuations when the sensor is exposed to wind are slightly but not significantly more erratic than the noise due to the sensor resolution. The data were collected using one of the custom-built devices [125] equipped with a barometer (BMP280, operating at about 1 Hz sampling rate). The device was placed behind a glass shield in an outdoor environment for the first hour and then exposed to the incoming wind by placing the device in front of the glass for the next 45 min.

Similarly to surface heating, the upper part of the atmosphere is heated differently over the period of a day. The resulting diurnal temperature variations give rise to a diurnal pressure cycle as illustrated in Figure 2b [117] with data collected with the custom-built device [125] (with BMP280, about 1 Hz sampling rate), which was placed on a desk in an indoor environment for 15 h (from 6:00 a.m. to 8:50 p.m.). This behavior has been well documented since the 1830s [118], with a semi-diurnal cycle with two peaks around 9 a.m.–12 p.m. and 9 p.m.–12 a.m., depending on the latitude [108]. In the tropics, the first late morning/afternoon cycle has the highest amplitude due to the maximum heating during the day, while the second night cycle has a lesser amplitude given the reduced solar heating [117]. In the mid-latitude regions, these peaks are shown to have similar amplitudes for both cycles [117]. The highest pressure variations are reported closer to the equator with 320 Pa between maxima and minima, while the mid-latitude difference does not exceed 80 Pa [117]. In addition to the periodical pressure variation, the absolute pressure also changes from day to day [108]. These changes are, however, prominent in mid-latitude regions (∼600 Pa on average), while they are smaller in the tropics (∼70 Pa on average).

Ho et al. analyzed pressure data from 2309 U.S. cities and observed that these variations are less than 100 Pa per hour for 99% of the time [78], while Liu et al. [69] observed that they could exceed 100 Pa for intervals larger than one hour and remained stable over short intervals of less than one minute. The time scale of the diurnal pressure cycle is hence much larger than many human activity time scales, and these slow variations are not concerning for applications related to study and classification of human behavior [28,100]. Ho et al. [78] used reference pressure from nearby weather stations to alleviate this effect when identifying the correct elevation of a vehicle path. Bao et al. [85] modeled the first-order difference of this pressure cycle as a white Gaussian stochastic process. Other studies have focused on using relative pressure changes called differential pressure instead of the absolute pressure [68,99,100].

### 4.2. Built Environment

Ambient pressure can be controlled in an enclosed space using mechanical systems like heating, ventilation, and air-conditioning (HVAC) systems [119]. Absolute and relative barometric pressure can thus be affected by these control systems. Buildings are positively pressurized compared to the exterior so as to have an outflow of air [120]. The magnitude of pressure differences is recommended by architects and regulatory authorities based on the function of the space. A minimum of 2.5 Pa is required for general living spaces, while a clean or aseptic isolation room is kept in a pressure difference of 12∼50 Pa to avoid contamination from the outside.

Lstiburek et al. [121] list the four types of pressure field experienced by a building: 1. exterior field—outside a building, 2. interior field—inside a building room, corridors, stairwells, etc., 3. interstitial field—building cavities, and 4. air conveyance system field—air supply, heating, exhaust systems, etc. The temperature, moisture, and pressure inside a building hence fluctuate over the day through the interaction of these pressure fields caused by the building structure, climate and weather, and the mechanical systems inside the structure [121]. Bollmeyer et al. [84] observed that temperature and humidity changes in a room have very little impact on barometric pressure. For instance, a temperature change of 10 degrees Celsius causes a ±1 Pa change, while a humidity change from 10% to 90% creates a pressure difference of less than 0.12 Pa.

Muralidharan et al. [68] observed that the type of building space (tall, short, narrow, and/or wide) and building pressurization also have little effect on barometric pressure measurement. They recorded pressure differences of less than 20 Pa even across multiple days. Xia et al. [66] similarly found no significant effects due to air conditioning in their prediction of floor location.

Even if HVAC systems and the built environment have little effect on HAR, the transition between indoors and outdoors has been shown to produce noticeable pressure changes. Wu et al. [74] found that a constant pressure difference of 25 Pa between the interior and exterior of a building creates a pressure difference of 20∼40 Pa measured by barometers integrated into smartphones when carried during a door opening. Similarly, Bollmeyer et al. [84] found a 30 Pa jump when a door is opened and a 20 Pa jump when a window is opened. Lstiburek et al. [121] showed in 2002 that indoor air conditioning can lead to pressure changes of approximately 2 Pa. More recently, Xu et al. [101] measured pressure differences between a room and the exterior of approximately 40 Pa.

### 4.3. Air Velocity during Motion

Vehicles like cars and buses are in quasi-equilibrium with their environment due to the vents and ducts allowing the airflow in. This means that the barometric pressure inside a vehicle is very close to the exterior pressure. Note that the term ‘exterior’ denotes the air pressure in the immediate vicinity of the vehicle and not the ambient pressure far from it, which could be drastically different. This is due to the fact that ambient pressure is increased by the vehicle’s motion near its surface where the dynamic pressure increases and the static pressure falls. The stagnation pressure on the surface caused by stopping the airflow near the surface produces a pressure distribution across the vehicle. This, in turn, creates pressure fluctuations inside the vehicle whenever its motion changes. However, this effect is transient, and a quasi-equilibrium is reached quickly between interior and exterior. In order to demonstrate the effect of air velocity on the barometric pressure, we collected the data with five custom-built devices [125] (BMP280, operating at ∼0.06 Hz sampling rate) in a bus and a car for ∼55 min and ∼30 min, respectively. Figure 3a,b illustrate this behavior when a barometer is carried by a human subject during a bus and car ride, respectively, where limited pressure fluctuations (∼50 Pa) are observed.

On the other hand, the transition between a stationary and moving vehicle can be quite noticeable. Ho et al. [78] found that the opening of a door/window during driving creates pressure changes of up to 30 Pa. During a car ride, they showed that switching the air conditioning on and off created a pressure difference of 50 Pa [78]. It is very challenging to attribute it to the climate-control system alone, as the air conditioning also brings in ventilated fresh air from the outside. However, Dimri et al. [77] observed that although there is a range of pressure jumps between different driving conditions (window open/closed, door open/closed, and air conditioning on/off), this did not affect the prediction of whether a vehicle was stationary or in motion. This shows that during vehicle motion, air velocity can substantially determinate the pressure jumps more than the vehicle environment itself. More importantly, the pressure fluctuations experienced by the vehicle during its motion are predominant. This is shown in Figure 3c (collected with a custom-built device [125], operating at ∼0.06 Hz sampling rate for ∼3 min in a subway train of the Downtown Singapore MRT line, from Upper Changi station to Jalan Besar station), where successive accelerations and decelerations of the train create significant pressure drops and rises, respectively, when it leaves and approaches a stop.

This effect can be severely amplified by the built environment, such as in tunnels. Vehicles passing through a tunnel experience a “piston effect”, where air is pulled inside the tunnel as the air is pushed back by the vehicle in motion [122]. This is shown in Figure 3b, where the car entering a tunnel leads to a drastic change of more than 200 Pa in the pressure measured inside the car. This effect has been shown to create a train of compression waves throughout the tunnel similar to sonic booms [123]. Sankaran et al. [79] showed that the pressure fluctuations during underground subway rides can exceed 200 Pa compared to a bus ride, which shows fluctuations of ∼50 Pa. Barnes et al. [124] analyzed road vehicle passage through tunnels in the Boston metropolitan area and found that the smaller the clearance between the vehicle top and the tunnel roof, the higher the negative pressure drop on the sensor fixed to the tunnel roof, measuring a drop of 100 Pa and 250 Pa for a clearance of 1.4 and 0.4 m respectively. During subway rides, we can clearly distinguish between a train stop and motion [79] (see also Figure 3c). By combining this behavior with relative elevation data of train stations, Hyuga et al. [106] used the pressure jumps to estimate the location of a user during a subway ride.

### 4.4. Altitude

Atmospheric pressure falls as we travel vertically upward above the Earth’s surface. This is due to the Earth’s gravitational pull of air molecules to the surface, which gives rise to a pressure gradient equal to −ρg, where ρ is the air density and *g* is the local acceleration of gravity. As a consequence, the rate of change of altitude with pressure is almost linear near the Earth’s surface, while it is almost exponential at higher altitudes when taking into account the variations of the air density with pressure and temperature [115]. This is again due to gravity, which is stronger near the Earth’s surface, combined with the fact that air molecules in lower atmosphere are compressed by the air molecules above them [127]. For all purposes involving human activity under 10 km altitude above sea-level, it is safe to assume a linear relationship with altitude that typically shows a pressure decrease of 115 Pa per 10 m climb [115].

To illustrate the magnitude of the effect in the context of human activities, we collected some data using two custom-built devices [125] and two mobile phones (Iphone 7 and Iphone XR) operating at 1 Hz in a building, using different vertical movement modes, including elevator rides, escalator rides, and stair-climbing. Figure 4 shows the barometric pressure change due to change in elevation through the three vertical movement modes. For the elevator ride (Figure 4a), we collected three types of movements between seven floors in a campus building (Singapore University of Technology and Design). First, the elevator was stopped at each individual floor, and the process was repeated twice (two descend and two climb). Then, the elevator was stopped once at the fourth floor between the ground and top floors; this was repeated six times. In the third part, the elevator moved directly from the top to the ground floor without any stop in between for ten times. For the escalator ride (Figure 4b), the data were collected using the same two-way escalators repeatedly and continuously for ten times in a subway train station (Upper Changi MRT station). The floor height difference of the escalator is about 7.3 m. For the stair-climbing (Figure 4c), the data were collected in three interconnected seven-floor buildings (same campus). The interconnections are made possible through bridges running between the third and fifth floors of the three buildings. The first three types of VDA—descent, climbing, and descent—happened within three floors of building 3, followed by walking to building 2 on the first floor, where stairs were climbed from the first to the the seventh (4:40 a.m.) and descended to fifth floor. Using the fifth-floor connection to building 1, the stair case to building 1 was reached and used to climb to the seventh and descend back to the first floor (4:46 a.m.). This was followed by climbing to the second floor of building 2 and descending to the first floor of building 2 using a series of smaller staircases (4:49 a.m.). After this, the following floor jumps were made in building 2: floor 1 to 2, 2 to 5, 5 to 1, 1 to 3, 3 to 1, and 1 to 3, with flat pressure readings indicating walks in corridors. The barometric pressure time series of the three types of VDA indicated that the change in pressure pattern has higher fluctuations as we move from the elevator ride, to the escalator ride, and to stair-climbing. This difference is because of the speed at which the vertical displacement took place and, in the case of stairs, it was affected by the staircase design (type, distance between consecutive stairs, etc.) and walking speed.

Pressure–altitude relationship: The pressure–altitude relationship can be derived from the fundamental equation for fluids at rest [126]. Assuming an incompressible fluid in isothermal conditions, the change in elevation is given by
(1)$$z_{2}-z_{1}=-\frac{p_{2}-p_{1}}{\gamma },$$
where γ=ρg is the specific weight of air with density ρ=1.225 kg/m${}^{3}$ and acceleration due to gravity g=9.81 m/s${}^{2}$ at standard sea-level conditions. This pressure–altitude elevation holds with negligible errors as long as the elevation under study is less than 10 km from sea level [126], which is the case in most studies.

### 4.5. Sensor Accuracy

The quality of measurement of barometric pressure is limited by the sensor’s accuracy. Both absolute and relative barometric pressure can change between devices due to differences in sensors and their characteristics. Here, the ensuing measurement errors are defined as being caused by such inherent limitations of the sensor and not due to other factors, such as the environmental ones discussed previously [72,85].

Device dependency: The device dependency is introduced to account for differences between devices and software platforms [66,99,100,108,113], manufacturing inconsistencies, and inappropriate calibration by the manufacturer [100]. Figure 5a illustrates how the time series of barometric pressure readings from two devices can differ. Using two custom-built devices and two mobile phones (Iphone 7 and Iphone XR) with the same mobile application (Physics Toolbox Sensor Suite App version 1.3.5), a human subject simultaneously carried the devices while climbing and descending on the same two-way escalators for ten times. The recorded barometric pressures between the two custom-built device data shown were significantly different, while the relative pressure was practically constant. Absolute barometric pressure thus needs to be calibrated between different devices for comparison [66]. This can be done before deployment or performed actively by using a reference pressure from nearby weather stations [85,108], buildings, or floor levels [66]. For instance, Ye et al. [99] used active peer-to-peer calibration when users detect each other and used the encounter network to calibrate all the devices. It is thus difficult or impossible to use barometers to measure absolute atmospheric pressure accurately without careful calibration; several studies have shown that it is possible to produce consistent relative pressure measurements [69,99,100,113]. The relative pressure, however, is also affected by the sensor’s resolution, drift, and noise.

Sensor resolution: The accuracy of the barometer is dependent on the built-in resolution of the sensor. Barometers embedded in mobile devices generally have a relative accuracy of ±10 Pa [66,99,104], while commercially available high-resolution sensors can reach an accuracy of ±1 Pa [72,76,85,90]. Sensor resolution is also affected by the measurement errors caused by noise. Haque et al. [72] used Allan Deviation (ADEV), a time-domain analysis, to estimate the non-stationary errors of four different barometer models, and listed the random noise processes that are dominant for a given observation period.

Sensor drift: Some sensors exhibit a drift in time due to faulty manufacturing or old age. Ho et al. [78] found this to be a temporary drift with a non-Gaussian distribution, and were able to remove it by modeling the noise as an Ohnstein–Uhlenbeck diffusion process—a process that pushes the drift towards its mean or center.

Sampling frequency: The recording frequency determines the completeness of the data. Weather stations generally send out data every hour, while mobile barometers embedded in mobile devices can be designed to output at a rate of 1∼20 Hz [76,79,82,83,92,94,104,114] or higher [67,80,89]. Depending on the activity to be recognized, this sampling frequency should be set appropriately to capture the actual time scale of the activity. As an illustration, Figure 5b shows the time series of barometric pressure recorded by two devices with different sampling rates—1 Hz and 0.062 Hz. A human subject carried the two devices using stairs in a seven-floor building. The shift of absolute pressure was caused by the different device issues. Although the overall patterns of the two time series were parallel, some details in the device with the sampling rate of ∼0.062 Hz (orange line) were missing in comparison to the barometric pressure recorded with the ∼1 Hz device. This indicated that altitude or human activity might not be accurately estimated or recognized during certain periods if the sampling rate is not high enough.

## 5. Discussion

In this review, we had set out to (1) describe sensor data collection processes to track human activities, both in general and more specifically in the case of barometers, (2) discuss the use of the barometer in human activity recognition and list its applications, and (3) understand the many factors that affect barometric pressure. This section reviews the important findings of this endeavor, explores the challenges for each application category, and recommends future research directions.

### 5.1. Key Findings

In Section 2, we have summarized the road map to sensor data collection by presenting its challenges and nuances. Unlike inertial or location sensors, barometers make use of the subtle changes in the ambient conditions to help track human activities. Depending on the application, barometers are also less dependent on the placement and orientation of the sensor, and of the physical characteristics of the subject themselves. They also require lower sampling rates and less pre-processing than inertial sensors. However, the recognition of activity classes using barometric sensor data faces similar issues to those of other sensors in terms of class imbalance, lack of long-term data in free-living environments, and accurate annotation.

In Section 3, we detailed the common methods used in different stages of activity recognition using barometers and their wide-ranging applications to track human activities. Since the late 1990s, the use of barometers has expanded beyond environmental sensing to tracking human activities. Measuring subtle changes in the ambient pressure has resulted in the ability to detect a vehicle’s movement and its location [77,78,79,106], human gait patterns [87], and vertical mobility [28,65,66,67,68,69,70], as well as to monitor indoor environments [74]. As listed in Table 1, the limited number of studies that have used only barometers are versatile in their applications, but have mostly considered experiments in specific environments (e.g., focusing only on indoors or on outdoors, on short-term or on long-term periods), limited classes of activities, and, more often than not, have failed to consider all the factors that can potentially affect the barometric pressure. The tracking of activities using barometric pressure data is complemented by geographical elevation maps, usage of multiple barometers, or other sources of barometric pressure data, such as reference stations. However, an accurate inference of a wide range of activities can be done by incorporating additional sensors’ data, such as inertial, location, or environmental sensors. In fact, these sensors can be important in removing or quantifying undesired factors that affect barometric pressure; for example, the use of GPS data can give the context of motion in vehicles, and the accelerometer can provide the state of motion or rest during ambulation. Table 2 showed the list of important studies that have used barometers as part of their sensor suite to improve the detection of vertical mobility and falls, as well as their estimation of energy expenditure in physical activity.

The studies listed in Table 1 and Table 2 highlight both the limitations of their experimental conditions and the lack of understanding of several factors that influence the measured ambient pressure. Section 4 hence endeavors to move in this direction by reviewing the fundamental properties of atmospheric pressure and inspecting their interaction with several environmental conditions that arise when a barometer is carried by a human. It also brings together a range of studies from different fields of science and engineering that have contributed to the improvement of the understanding of the factors influencing atmospheric and barometric pressure and to the quantification of their respective magnitudes (see Figure 1). All weather events that have happened on our planet have been observed to produce less than 5050 Pa change in barometric pressure [116]. While a high-speed wind can cause a change in pressure in the order of 100 Pa [126] in a matter of seconds, many of the changes by climate and weather factors happen over a longer time scale, such as the one due to diurnal pressure cycle (Section 4.1). Climate-controlled buildings similarly cause changes in the measured ambient pressure during the transition between indoors and outdoors, and when opening and closing doors or windows (Section 4.2). Even though this causes a transient change in barometric pressure measurement, these changes are observed to be less than 40 Pa [74,84,101,121]. During vehicle motion, the dynamic pressure in the immediate vicinity of the vehicle increases and decreases significantly during acceleration and deceleration, respectively (Section 4.3). This produces a transient change in pressure of 50∼300 Pa in a matter of seconds [79,124]. This change is magnified by the presence of built environment like tunnels and bridges. In addition, the sensor itself can be a source of error. For example, the difference in measurement of absolute pressure between devices, the inherent resolution of the sensors, sensor drift over a long time, and the sampling frequency of the collected data can all add to the noise in the measurement (Section 4.5). The approximately linear relationship between barometric pressure and altitude for less than 10 km from sea level is hence affected by all these factors [126]. Any reasonably long-term tracking of individuals using a barometer should hence be aware of these factors to interpret the collected data accurately.

### 5.2. Challenges and Future Directions for Different Applications

The various factors that influence barometric pressure have been used by many studies to make better modeling decisions to accurately identify the activity class of interest. The steps to understand these factors have recently led to creative use of the barometric pressure measurements to identify some classes of activities that are traditionally detected using other sensors. To review the challenges and recommend future directions for different applications of barometer data, we categorized the studies that track human activities into the following groups—(1) human mobility: tracking movement of people, (2) health monitoring: tracking elderly, patients, or healthy people for health-related activities, (3) vehicle tracking: tracking movement of people riding vehicles such as cars, buses, trains, etc., and (4) building monitoring: monitoring the movement of people through the changes in building environments. A taxonomy of barometer applications to track human activities is shown in Table 4.

Human mobility: The subtle changes in barometric pressure during body movements are used to detect walking, understand gait patterns, and count steps [87]. Similarly to the case of accelerometers, which are typically used in this application, this requires a high sampling rate (∼5 Hz) for barometers (albeit less than the one needed for accelerometers, ∼20 Hz). This tracking also requires a high sensor resolution (±1 Pa) that is usually not present in commonly available devices, like mobile phones (±12 Pa) [28]. Hence, future tracking of walking and gait patterns requires custom-built devices that have a high resolution and sampling rate, or the wearable industry should move in the direction of assembling such high-performance barometers in ubiquitously available devices. However, this will also lead to higher battery consumption.

In the field of indoor navigation and positioning [28,65,66,67,68,69,70,71,72,73], vertical displacement activity can be tracked with barometer data. One important challenge in tracking VDA accurately is the need for a reference pressure to keep track of the floor changes. Another is the lack of accounting for factors—other than altitude—that affect barometric pressure in indoor environments, such as the indoor-to-outdoor transition due to climate-controlled buildings, diurnal pressure cycle that slowly drifts the reference pressure, and difference in absolute pressure measurements that necessitates an active calibration between devices. When aiming at the detection of VDA, barometric signal data need to be considered not only to recognize altitude changes, but also to determine the mode of vertical transport (stair-climbing, slope, escalator or elevator rides). In studying epidemic disease propagation, for example, it makes a significant difference to recognize whether the subject is in an elevator (closed space) or on an escalator (open space). The identification of vertical transportation types could be recognized through some pattern recognition of the experimental barometric pressures. There are a lot of useful extensions of barometer sensor applications in various fields that require further exploration.

Understanding human mobility has far-reaching applications in urban planning and economic development [128]. While traditional study of mobility used GPS or other location-based sensors [129], studying vertical mobility requires the use of barometers. The derived vertical displacement from such studies and subsequent statistics of vertical displacements can help better model and predict mobility in a building or at the city scale. The variation of the natural environmental conditions during mobility over the course of the day leads to a complex interplay of factors that influence the barometric pressure. Hence, the application of studying city-scale human mobility in its vertical dimension demands a deeper understanding of all these factors.

Health monitoring: Currently available wearable devices are increasingly designed to track health-related activities. In healthy populations, physical exercises like running, walking, playing, and the rest time between them are estimated with the help of a suite of sensors, including barometers. Barometers have been known to improve the tracking of physical activity, resulting in a better estimation of energy expenditure [44,91,97,112]. While barometers have been used to identify walking or to count the number of steps [87], their use in tracking other physical activities (e.g., running) has not been explored. To track the elderly or patients with health issues, barometers are primarily used to detect falls from wheelchairs, bed, or stairs [76,83,93]. It is important to note that this requires high-resolution barometers whose errors should be smaller than the height of the potential fall.

Vehicle tracking: As mentioned in Section 3.4, the topographical features in our transportation routes can be used to track vehicles. An important challenge here is that air velocity during motion (Section 4.3) can lead to changes in pressure as high as 300 Pa, which can result in an altitude estimation error of 25 meters (12 Pa ≃ 1 m). On the other hand, the change in pressure due to this factor can itself be used to distinguish the vehicle ride from vehicle stops. The identification of vehicle stops, where there are no pressure jumps, has been used successfully as the point to reliably convert the barometric pressure into altitude [106]. A better understanding of the relationship between the barometric pressure and the type of vehicle, velocity of travel, and configurations of the vehicle (e.g., opened/closed windows) could lead to better models and predictions. Moreover, similarly to tracking VDA over long time, a reference pressure from local stations and active calibration between devices are necessary to accurately keep track of the elevation changes during vehicle rides.

One important application of tracking vehicles is to locate and track the users in lieu of using GPS [44,77] (or other location sensors), which are known to be energy expensive. While this requires an initial location input, reference pressure, active calibration, geographic elevation map, and tracking of vehicle stops (or slowing down), the primary limitations is that it might work only in routes with measurable change in elevation. It is also worth noting that in the absence of an efficient algorithm to match the barometric pressure changes to possible routes using the elevation map, using a barometer might be as consume as much energy as using a GPS. Incidentally, in mountainous regions where mobile network connectivity is usually sparse, this application may particularly find its niche to supplement GPS.

Building monitoring: As mentioned in Section 4.2, buildings with HVAC systems have a dynamic airflow environment where barometric pressure change is noticeable during indoor-to-outdoor transitions and opening/closing of doors or windows. While there are applications for building security [74], this also enables tracking of human movements. For example, the identification of indoor-to-outdoor transitions has been used to begin the calibration of devices entering a building to keep track of floor changes. Inside the buildings, the knowledge of different room pressurization or conditions of open/semi-open spaces can be used to determine the location of the user and help improve the indoor localization and navigation systems in a building. This naturally requires barometers to be carried by the users and installed in the environment. Similarly to other applications, detecting changes in pressure that are typically <40 Pa in indoor environments requires barometers with a high sensor resolution (±1 pa) and sampling rate (∼5 Hz), which are not available in mobile devices. Hence, an array of barometric sensors to monitor a building can lead to an accurate tracking of human movements in indoor environments.

Furthermore, barometers are increasingly used as a practical complement to other sensors for tracking human activities. As noted before, a multi-sensor setup could provide more flexibility in the recognition of activities and more comprehensive information on the environment of the sensors and carriers. However, a multi-sensory data collection setup would generate a large variety in collected data, leading to challenges in data integration. Methods that enable fusion of sensor data of barometric signals to other sensor data need to evolve to fully realize the barometers’ full potential.

Table 1 and Table 2 also show that for many applications, not all the factors that can influence pressure are always considered, even after taking into consideration the limited scope of these studies. These factors are not always fully understood, and their magnitude is not universally agreed upon due to the wide range of conditions under which these experiments are performed (see Figure 1). It is thus crucial to understand and quantify all the factors affecting the barometric pressure when working on a particular problem or application related to human activity tracking and recognition. Some of these effects may be irrelevant, while others might impair our ability to properly identify patterns of activity from the sensed data.

### 5.3. Privacy Issues

Tracking human activities inevitably leads to privacy issues. Anonymity of individuals cannot be guaranteed even with coarse spatial and temporal resolutions of the collected data [130]. Re-identification of individuals from seemingly anonymous data has been shown to be effectively performed due to unique signatures of human movement [130,131]. This issue arises as well with the use of barometers, where re-identification of human subjects could be possible using the tracking of altitude changes during the start and end of each day. In effect, a barometer can be used to perform the so-called last-mile tracking that increases the spatial resolution from a block to a floor. This is a particularly pronounced problem in highly vertical cities, and a proper framework to protect the privacy should take into account the potential use of barometer data when tracking human activity. Indeed, privacy issues are always a major challenge for all human activity analysis or social-related studies. Previous research introduced a series of geo-masking techniques for the improvement of privacy protection [132,133,134,135], but most of these techniques could only be used for 2D spatial data, leaving a large gap for the human activities in the vertical dimension.

## 6. Conclusions

Today, barometers are found in almost (if not all) wearable devices and smartphones. The vast breadth of applications listed in this review underscores the tremendous potential for use in extracting barometric pressure data on a massive scale. Some attempts with limited scope are reported in Table 1 and Table 2. However, the systematic use of barometer data could be envisioned on a large scale and on long time scales as a tool to study some aspects of the behavior of large populations of humans (or other living creatures) in their free-living environments, similarly to what has been done with mobile phone call detail records for studies of communications and movement.

Here, we reviewed and discussed a series of previous studies on using barometers in the tracking and recognition of human activity and movement patterns. Although numerous factors may affect the barometric pressure, the potential of using barometers as sensors for tracking human activity is enormous and worth the anticipation. While most previous studies focused on short-term, indoor, and small-scale human activities, currently, there is a large research gap in the utilization of barometers for a study on a large spatial scale (e.g., city-wide or on a national level) and on a long temporal scale for human activity tracking, recognition, and analysis, specifically concerning the detection of vertical movements.

## Figures and Tables

**Figure 1 sensors-20-06786-f001:**
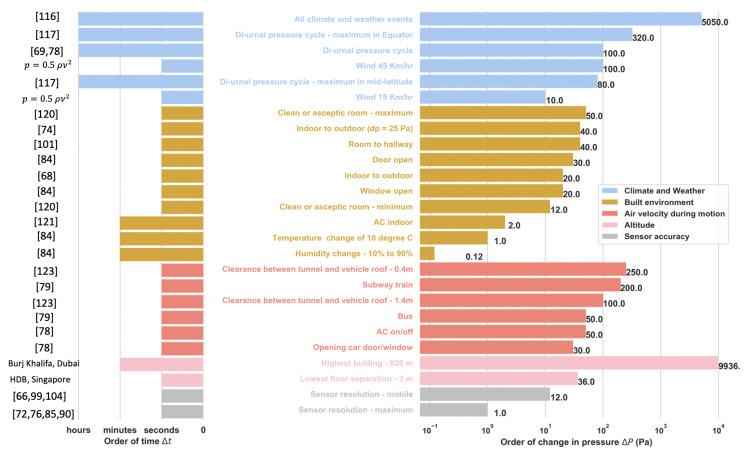
Orders of magnitude of changes in pressure and of the corresponding timescales for several factors influencing barometric pressure.

**Figure 2 sensors-20-06786-f002:**
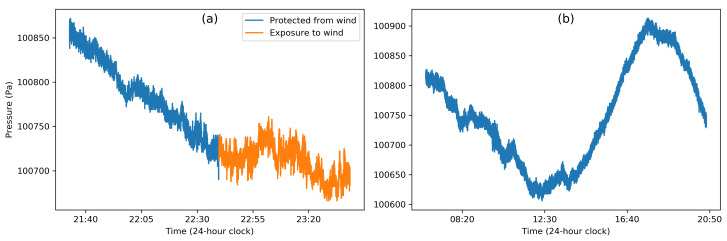
An illustration of the effects of climate and weather on barometric pressure: (**a**) effect of wind and (**b**) diurnal pressure cycle in Singapore. Recorded by a custom-made device (barometer model: BMP280) with ∼1 Hz sampling rate.

**Figure 3 sensors-20-06786-f003:**
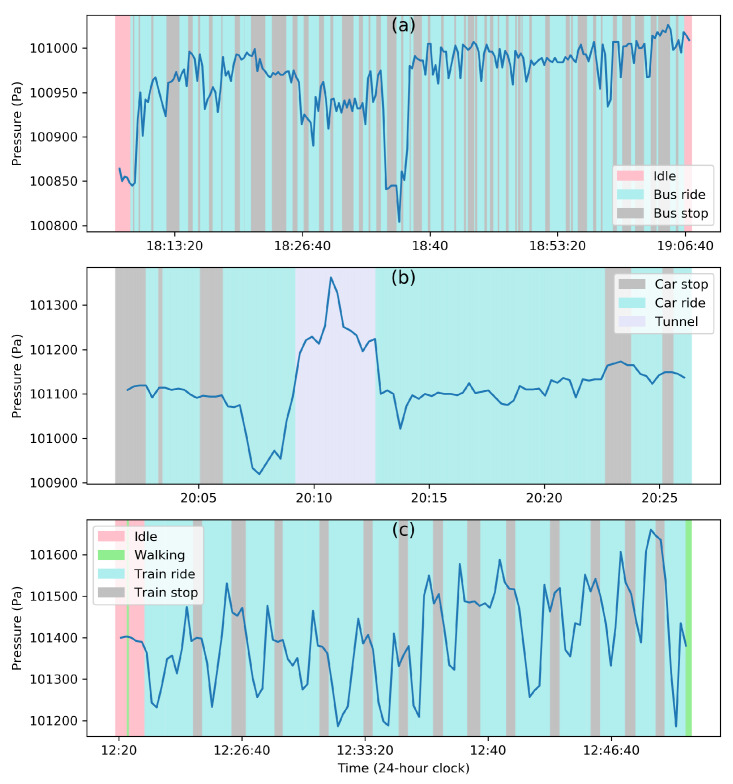
Illustration of the effect of air velocity during motion for a (**a**) bus ride, (**b**) car ride and the effect of the built environment (tunnel), and (**c**) subway train ride. Each panel shows the effect of air velocity during motion on the barometric pressure, with different modes of transport showing different types of changes. Subway train rides yield the highest changes in magnitude (∼200 Pa), while bus and car rides show relatively smaller amplitude changes (∼50 Pa), except when a car is entering a tunnel. The panels also show the effect of elevation changes during travel. In panel (**a**), there is a fall and rise in pressure around 18:35, corresponding to a climb and descent on the road path. Similarly, a drop and rise in pressure is observed in panel (**b**), where the car climbed and descended on an elevated bridge just before entering a tunnel. In panel (**c**), the acceleration of the train as it leaves the stop creates a sudden pressure drop, and a rise in pressure is subsequently observed as it approaches a stop, followed by a small dip in pressure as the train comes to equilibrium with the station environment, thus creating a repeating pressure pattern. Data were recorded by carrying a custom-made device (barometer model: BMP280) in different transport modes with a sampling rate of ∼0.06 Hz.

**Figure 4 sensors-20-06786-f004:**
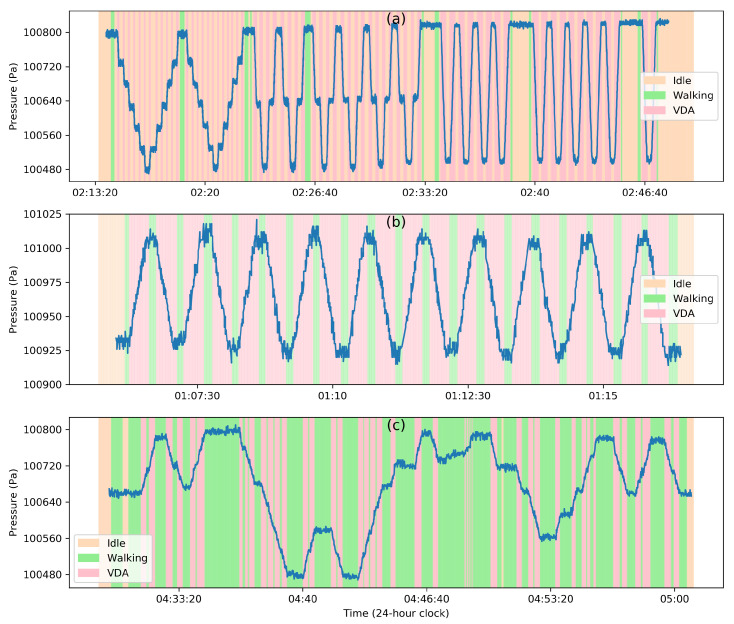
Effects of variations of altitude during human activity and motion on (**a**) an elevator, (**b**) an escalator, and (**c**) stairs. Data were collected by carrying a mobile device with a sampling rate of ∼1 Hz on different modes of vertical movement. For panel (**a**), the elevator was intentionally stopped at each floor in the first part of the data collection (before 2:23 a.m.) to show the ability to distinguish single-floor changes. In the second part (between 2:23 a.m. and 2:33 a.m.), only one stop was made between the first and last floor. Finally, the last part of the data (after 2:33 a.m.) corresponds to an uninterrupted elevator ride between the first and last floor. The data in panel (**b**) were collected by continuously climbing and descending on the same two-way escalator, and the data in panel (**c**) were collected while using stairs. The escalator climb and descent in panel (**b**) correspond to an average relative pressure difference of 80.7 Pa with a standard deviation of 3 Pa, thus showing the accuracy in recording relative pressure changes. Moreover, the relative pressure of 80 Pa corresponds to a height of 6.6 meters according to Equation (Equation 1), thus close to the measured height of 7.3 m and within the equivalence sensor resolution of ±1 meter. The data in panel (**c**) were collected by a human subject performing a series of floor jumps (climbing, descending, and walking) in three interconnected campus buildings: floor 3 to 1 (4:32 a.m.), 1 to 3, 3 to 1, 1 to 7 (4:40 a.m.), 7 to 5, 5 to 7, 7 to 1 (4:46 a.m.), 1 to 2, 2 to 1 (smaller staircases, 4:49 a.m.), 1to 2, 2 to 5, 5 to 1 (4:56 a.m.), 1 to 3, 3 to 1 (4:59 a.m.), and 1 to 3. The barometric pressure fluctuations are highest for the stairs due to the low speed of vertical displacement, difference in staircase designs, and inconsistent walking speed.

**Figure 5 sensors-20-06786-f005:**
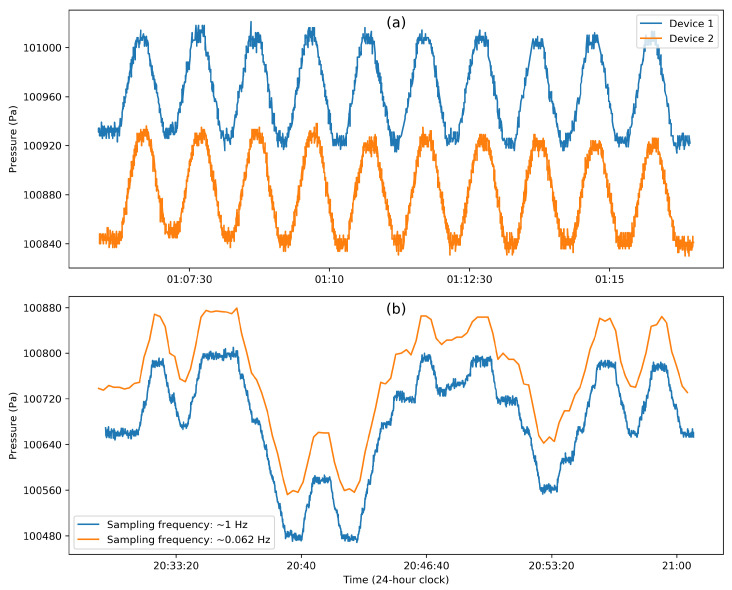
The effects of sensor accuracy. (**a**) Device dependency: Two custom-built devices were simultaneously carried by a human subject on an escalator to record the barometric pressure at a ∼1 Hz sampling rate. The absolute pressure measured by each device is significantly different, while the relative pressure is practically constant. (**b**) Effect of sampling frequency: Two custom-built devices were simultaneously carried by a human subject on stairs. They were both embedded with the same microelectromechanical system (MEMS) barometer model (BMP280), but with different sampling rates, i.e., ∼1 Hz and ∼0.062 Hz, respectively. This panel shows how the sampling rate affects the detection of altitude changes.

**Table 4 sensors-20-06786-t004:** Taxonomy of barometer applications to track human activities.

Application	Description	Reference
Human mobility	Tracking movement of people	[28,67,68,69,84,87,101,104]
Health monitoring	Tracking elderly, patients, or healthy people for health-related activities.	[44,75,76,83,91,93,95,97,112,112]
Vehicle tracking	Tracking movement of people riding vehicles such as cars, buses, trains, etc.	[77,78,79,106]
Building monitoring	Monitoring the movement of people through the changes in building environments.	[66,68,71,74,84,99,108,121]

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
