# Peer review of "On the Challenges and Potential of Using Barometric Sensors to Track Human Activity"

_sensors, 2020, doi:10.3390/s20236786_

Round 1

Reviewer 1 Report

This article presents an extensive bibliographic review covering relevant points for track human activity, which is a very interesting topic. However, in order to clarify some points, it is advisable to analyze and respond the following points:

  1. About the purpose and content of the article, it is not clear, and it is necessary to distinguish, since in the abstract it says:

Page 1, line 9: “This review article thoroughly details all these factors and presents a comprehensive report of the numerous studies dealing with one or more of these factors, in the particular framework of human activity tracking and recognition”.

On the other hand, in the description of the content, shown on page 2, line 114:

About section 4: “This section is enriched by data especially collected for illustration purposes".

Therefore, it is necessary to adapt the summary, where in addition, a little phrase about relevant conclusions obtained from collected data could be added.

  1. If data are presented from measurements performed by authors, it is necessary to show an adequate and clear methodological description or subsection, IN EACH CASE (Figures 2-5), in the corresponding place in the body text. This information is currently briefly shown, in some cases, the way of collecting the information is described in little detail as part of the description of the mentioned Figures, located at their footer. In addition, it is necessary to be more specific about the conditions under which some of the measurements and equipment used are made, since for example, in some cases, only the sensor model is mentioned.

Also, it is restricted to a single measurement, except Fig. 5, where 2 mobile phones were used, so the repeatability is not clear, it is suggested to emphasize the illustrative purpose.

  1. On page 6, line 234 “hoz” must be “Hoz”.
  2. Figure 1 (at the beginning of the page 14) is shown without previous reference in the text, thus make difficult its interpretation. This figure is cited until the Future Research section. It is necessary to modify or order the content, in the proper way.
  3. Conclusions section is absent. Especially, when making measurements, it is to be expected that it exists.
  4. Two references are about Wikipedia; it is suggested to replace by more formal bibliography.

Author Response

See PDF file.

Reviewer 2 Report

In the article “On the use of barometric sensors to track human activity”, the authors thoroughly review the barometer sensor characteristics and its use in studies related to human activity tracking and recognition. The article reads well.

The introduction is well written.

Line 61- 62: I suggest the authors mention activities of healthy and diseased. The symptomatic movements for different movement disorders deviate from classical human activity.

Line 97:  While talking of vertical displacements, it would be beneficial to talk of any applications in sleep?

It would be good to see comparisons of barometric data with accelerometric data in terms of detection and accuracy of gait.

Author Response

See PDF file

Reviewer 3 Report

The article reviews the relation between barometer parameters and the human activities tracking. After giving a rough overview of barometer sensor development history and the general issues in sensor applications, the author summarizes its applications in human activity recognition. This review lists those parameters that may affect the accuracy in human activity recognition, such as climate and weather, built environment, air pressure change during motion, altitude change, intrinsic sensor accuracy. The material present in the review is comprehensive and suitable for publication. However, since this review is mainly focused on reporting (summarizing) the issues appears in using the barometer for human activities, but not its achievement or algorithm development. I suggest revising the title as “On the issues of barometer sensor in tracking human activities”.   

Author Response

See PDF file

Reviewer 4 Report

The paper aims to present a review on the use of barometers for human activity recognition and identify and discuss factors affecting the atmospheric pressure and sensor accuracy.

The paper's focus area is novel and interesting. It is well written and lots of good references included.

However, there are issues that need to be addressed.

The paper claims that its review focus is on the use of barometers to track human activities but most of the materials are about the history, physics and hardware of a barometer and very little about its use for activity recognition. Mainly Table 1 has the review of applications with insufficient discussions.

The introduction starts with a great deal of discussion about the history of barometer. This information can be shortened and moved to other parts. Instead, the introduction should provide a high level overview of this survey focus areas (and why) and clearly specify what are the objectives of this survey.

The paper doesn't have a conclusion section. This should be added.

The current Future Research section does not really talk about future plans but is a mix of everything, and some of findings from the table.

The paper must have a section (before Conclusion) called Discussion or Key findings where it discusses the main findings from this review that matches the objectives mentioned in the introduction, what are the principal findings (could be related to Table 1) and from other parts, what are the open issues, and what is the roadmap for future research. This main part is missing at the moment, that is the most important part of any survey.

The structure and presentation of materials also require revision.

Section 2 title needs revision, what 'general' means here? is this section only about data collection?

The subsections under 2.0 are very short, some are just one short paragraph talking about several things, so is not clear what message each subsection tries to convey. These subsections need revision (with a clear focus) and either merged or extended or using a different structure.

The section 3 title is not consistent with Section 2 (one has 'tracking' and one has 'recognition' and section 3 adds 'mobility' too what wasn't used before.

Section 3 has two long subsections (barometer only and multi-sensor) without numbered headings (compared to the short numbered headings in Section 2). These need to change to numbered headings.

Section 3 has Applications and a table of related papers. Since they include both barometer only and multi-sensor studies, it is better to be broken into two tables and each one can be included in its own subsection and findings discussed separately.

The criteria for comparison and analysis of works in Table 1 are not defined and justified before the table. There is a need for a following section after the table such as the Results section where the findings from the table based on the criteria can be discussed.

At the moment, only one of the table criteria (Factors considered influencing
barometric pressure) is presented and discussed after the table without any explanation. It is very confusing.

Another main thing that is currently missing from the applications' review is categorization of these applications and how each group uses barometer and what are the challenges for each category. Based on the title and aim of this paper, this information is the most important part and is missing now. Only under Future Research there is some reference to these categories "gait patterns, step count, environmental monitoring, building monitoring, vehicle tracking, and health monitoring'. This part also needs to be added.

Author Response

See PDF file

Reviewer 5 Report

In this work, the authors present a literature review on the use of barometers in the scope of Human Activity Recognition (HAR). The authors present several application examples were the barometers were employed, describe several stages of the sensor data collection process, and discuss some factors that influence the performance of barometers. Overall, the work presented is quite solid and worth of publication. Although the scope is quite narrow (the use of barometers for HAR), it is an interesting and current research topic. Nevertheless, there are a few topics that the authors should consider before the paper is ready to publish. These topics are:

  1. overall the paper is quite easy to follow and read. However, some parts of the introduction are a bit overcomplicated to follow. The authors should review the introduction and add some references in some key sentences (e.g., p.2 l.47-49). Some sentences/ideas seem to be unfinished;
  2. Section 2 is the weaker part of the document. The authors state that their focus is on the use of barometers but most of section 2 describes problems that happen with other sensors. For some subsections, only one or two lines are targeted to the barometers and, in most cases, are too generic. I would like the authors to rewrite this section and focus on the specific case of barometers;
  3. Section 3. In the abstract and introduction, the authors state that their focus is on the use of barometers for HAR. However, some applications described in section 3 do not target for HAR. I would like to recommend the authors to remove the works that do not focus on HAR;
  4. Finally, a taxonomy with the several applications in the context of HAR that rely on barometers should be created to highlight the importance of such sensor.

Author Response

See PDF file

Round 2

Reviewer 1 Report

There are not more suggestions.

The authors of this review have attended to all my reccomendations. 

Many issues have been considered, such as the effectiveness of barometers in the field of human activity, considering numerous factors affecting the atmospheric pressure, as well as to the properties of the sensors under analysis.

The measurements are now adequately described.

Reviewer 4 Report

Authors have done a great job addressing all the comments and the paper has significantly improved and now its contributions to this field are evident.

Reviewer 5 Report

All my comments were properly addressed and, in my opinion, the current version of the work is worthy publication. Congratulations to the authors for the good work.